# The effect of rehabilitation interventions on physical function and immobility-related complications in severe stroke: a systematic review

Mark P McGlinchey [1,2] Jimmy James,[2] Christopher McKevitt,[1] Abdel Douiri,[1] Catherine Sackley[1]

¹Department of Population Health Sciences, School of Population Health and Environmental Sciences, King's College School, London, UK
²Physiotherapy Department, Guy's and Saint Thomas' NHS Foundation Trust, London, UK

**Correspondence to**
Mark P McGlinchey;
mark.p.mcglinchey@kcl.ac.uk

## ABSTRACT

**Objective** To evaluate the effectiveness of rehabilitation interventions on physical function and immobility-related complications in severe stroke.

**Design** Systematic review of electronic databases (Medline, Excerpta Medica database, Cumulative Index to Nursing and Allied Health Literature, Allied and Complementary Medicine Database, Physiotherapy Evidence Database, Database of Research in Stroke, Cochrane Central Register of Controlled Trials) searched between January 1987 and November 2018.

**Methods** The Preferred Reporting Items for Systematic Reviews and Meta-Analysis statement guided the review. Randomised controlled trials comparing the effect of one type of rehabilitation intervention to another intervention, usual care or no intervention on physical function and immobility-related complications for patients with severe stroke were included. Studies that recruited participants with all levels of stroke severity were included only if subgroup analysis based on stroke severity was performed. Two reviewers screened search results, selected studies using predefined selection criteria, extracted data and assessed risk of bias for selected studies using piloted proformas. Marked heterogeneity prevented meta-analysis and a descriptive review was performed. The Grading of Recommendations Assessment, Development and Evaluation approach was used to assess evidence strength.

**Results** 28 studies (n=2677, mean age 72.7 years, 49.3% males) were included in the review. 24 studies were rated low or very low quality due to high risk of bias and small sample sizes. There was high-quality evidence that very early mobilisation (ie, mobilisation with 24 hours poststroke) and occupational therapy in care homes were no more effective than usual care. There was moderate quality evidence supporting short-term benefits of wrist and finger neuromuscular electrical stimulation in improving wrist extensor and grip strength, additional upper limb training on improving upper limb function and additional lower limb training on improving upper limb function, independence in activities of daily living, gait speed and gait independence.

**Conclusions** There is a paucity of high-quality evidence to support the use of rehabilitation interventions to improve physical function and reduce immobility-related complications after severe stroke. Future research

### Strengths and limitations of this study

► This is the first systematic review to investigate rehabilitation interventions specifically to survivors of severe stroke.
► The review included outcomes on physical function and immobility-related poststroke complications, of which the latter contribute to high levels of caregiver burden and are less commonly reported outcomes in stroke rehabilitation research.
► Marked heterogeneity of included studies prevented meta-analysis.
► Most included studies were rated as low or very low-quality evidence due to unclear or high risk of bias as well as recruitment of very small samples.

investigating more commonly used rehabilitation interventions, particularly to reduce poststroke complications, is required.

**PROSPERO registration number** CRD42017077737

## INTRODUCTION

Despite advances in stroke management over recent decades, stroke remains one of the most common causes of death and disability globally.[1 2] The mainstay of treating stroke is stroke rehabilitation, which aims to enable a person to achieve their optimal physical, cognitive, communicative, emotional and social level of function.[3–5] Rehabilitation of physical function comprises a large component of stroke rehabilitation programmes delivered by healthcare professionals, such as physiotherapists and occupational therapists.[6–8] While several systematic reviews support the use of rehabilitation interventions to improve aspects of physical function, such as motor function, balance, walking speed and activities of daily living (ADLs),[9–11] it is not clear from these reviews if these

interventions are effective for survivors of differing levels of stroke severity, particularly severe stroke.

Severe stroke can be understood as a stroke resulting in a significant amount of brain tissue damage and multiple neurological impairments, which leads to a significant loss of function and residual disability.[12] Dependent on how it is measured, 14%–31% of people who sustain a stroke globally are classified as having a severe stroke,[13–18] a cohort of the stroke population that experiences worse outcomes compared with survivors of less severe stroke.[19–30] In the initial hospitalisation phase poststroke, they are more likely to develop acute medical complications, which are negatively associated with functional recovery.[19] Three month mortality can be as high as 40%, compared with just under 5% for those patients with mild stroke.[20–22] Survivors of severe stroke spend longer in hospital, resulting in increased hospital costs, and demonstrate slower and less functional recovery, resulting in greater dependency when they are discharged from hospital.[14 15 23 25] For those discharged from hospital, survivors of severe stroke are at least eight times more likely to be discharged to a nursing home.[25 26] Longer-term care costs, which mostly support survivors of severe stroke, represent 49% of total stroke care spending globally.[27] In the first year post severe stroke, mortality can be as high as 60%[20] and survivors of severe stroke also experience very high levels of immobility-related complications, such as falls, contracture, pain and pressure sores.[28 29] Due to this residual disability, the physical assistance provided by caregivers to look after survivors of severe stroke as well as the psychosocial and emotional impacts of the stroke on caregivers result in high levels of caregiver burden.[30]

As there are a number of significant issues faced by survivors of severe stroke, rehabilitation of severe stroke should focus on addressing these poor outcomes, particularly reduced physical function and its associated complications. However, the extent to which rehabilitation can address these outcomes is not clear. A previous systematic review demonstrated positive benefits of inpatient stroke rehabilitation, such as reduced mortality and hospital length of stay, and uncertain benefit on improving functional recovery.[31] However, this review did not explore the effect of specific interventions delivered within inpatient rehabilitation on improving physical function or on reducing immobility-related complications. Most trials investigating the efficacy of rehabilitation interventions on physical function have either not recruited survivors of severe stroke or not reported results specifically for survivors of severe stroke.[9–11] Therefore, it is not known if research findings are applicable to survivors of severe stroke. It is not clear whether rehabilitation should focus more on functional restoration, which may be incomplete or not possible, or reducing immobility-related complications, which may lessen longer-term burden for caregivers of severe stroke survivors. Due to this lack of clarity, there is an urgent need to summarise evidence-based rehabilitation interventions designed to optimise

physical function and reduce immobility-related complications for this cohort of the stroke population.

This systematic review aims to establish the effectiveness of rehabilitation interventions on physical function and immobility-related complications for survivors of severe stroke and identify areas for future rehabilitation research for these patients.

## METHODS

The systematic review has been reported according to the Preferred Reporting Items for Systematic Reviews and Meta-Analysis statement (see online supplementary file 1).[32] The protocol for the systematic review has been published previously.[33]

### Study design

The systematic review included randomised controlled trials (RCTs). The systematic review excluded quasi-experimental, correlational and descriptive study designs. Studies were selected according to the participant, intervention, comparator and outcome (PICO) format. The systematic review protocol provides full details of the PICO components[33] and a brief summary of the components is reported below. There were no deviations from the protocol PICO.

### Participants

The review included studies of adult (≥18 years) stroke patients with severe stroke. Stroke severity was defined using a score on a validated and routinely used outcome measure (eg, National Institutes of Health Stroke Scale (NIHSS), Functional Independence Measure (FIM), Barthel Index (BI)).[34–36]

### Interventions

The review included studies that involved the provision of rehabilitation interventions used to manage problems relating to physical function or immobility-related complications poststroke. A rehabilitation intervention was defined as any non-surgical or non-pharmacological intervention used in current clinical practice as part of the usual rehabilitative care of stroke patients.

### Comparators

The review included studies that had a comparator, which included any of the following: another type of rehabilitation intervention, usual care or no intervention. Usual care was defined as the rehabilitation that the patient would normally receive as part of undergoing stroke rehabilitation.

### Outcomes

The review included studies that focused on the primary outcomes of physical function and poststroke complications. As per the definition of function in the International Classification of Functioning, Disability and Health, physical function was assessed using measures of body function (eg, Fugl-Meyer Assessment), activity

(eg, BI) and participation (eg, Stroke Impact Scale).[37 38] An immobility-related complication was defined as any medical problem arising after a stroke because of immobility or reduced physical activity.[39]

### Search strategy

#### Information sources

Electronic searches of the following databases were conducted: MEDLINE, EMBASE, Cumulative Index to Nursing and Allied Health Literature, Allied and Complementary Medicine Database, Physiotherapy Evidence Database, Database of Research in Stroke and the Cochrane Central Register of Controlled Trials. An example search strategy is shown in online supplementary file 2. Databases were searched from January 1987 to November 2018. The search timeframe was guided by a scoping review of the literature (demonstrating very few published RCTs before 2000) and a consideration to include studies reflecting current clinical practice. Ongoing studies were identified by searching the Stroke Trials Registry (www.strokecenter.org/trials/) and clinicaltrials.gov. These sources were searched from 2012 to 2018 as it was assumed that studies before these dates would have been completed and published. References from included studies were hand searched and any potentially relevant study was included for review. Forward citation checks of included studies were also performed. To avoid language or cultural bias, studies in any language or geographical location were included.

### Data management and study selection

The results from the literature search were uploaded to a reference management programme (Refworks) and duplicate references were removed. A final list of non-duplicated references was generated by one author (MM). The titles and abstracts of the search results were screened independently by two review authors (MM and JJ) and full text articles were obtained for relevant studies. Full text articles were reviewed by the same two authors (MM and JJ) independently to determine if studies met the inclusion criteria using an inclusion/exclusion checklist previously piloted. Two review authors (MM and JJ) independently performed data extraction for all eligible articles using a data extraction proforma previously piloted. Any differences in opinion between the two authors at any stage of the study selection and data extraction process were resolved by a third review author (CS).

### Risk of bias and quality assessment

Risk of bias was assessed by two review authors independently (MM and JJ) using the Cochrane Collaboration tool for assessing the risk of bias across six main domains (sequence generation, allocation concealment, blinding, incomplete outcome data, selective outcome reporting, other bias).[40] A risk of bias judgement of 'high', 'low' or 'unclear' was determined for each of these main domains. The strength of evidence was assessed using the Grading of Recommendations Assessment, Development and Evaluation (GRADE) approach.[40] The five criteria considered by the GRADE approach included risk of bias, inconsistencies between studies, indirectness, imprecision and publication bias. Studies were given a baseline rating of 'high' and downgraded if any of the five criteria were present. The quality of the evidence was ranked 'high', 'medium', 'low' or 'very low' by two review authors independently (MM and JJ). Any differences in opinion between the two authors at any stage of the study selection and data extraction process were resolved by a third reviewer (CS).

### Data analysis

Due to the limited number of studies investigating each individual intervention and the marked heterogeneity of the selected studies, it was not appropriate to undertake a meta-analysis. Heterogeneity was seen in the rehabilitation interventions (type, dosage, method of delivery, timeframe completed poststroke) as well as outcomes (type and timeframe completed poststroke). Therefore, a descriptive review of results was performed. As there may be differences in recovery rates and outcomes according to the time poststroke, studies were grouped into three timeframes poststroke based on when participants were recruited to the study and when the study finished. These timeframes were the acute to early subacute stage (up to 3 months poststroke), acute to late subacute stage (up to 6 months poststroke) and chronic stage (greater than 6 months poststroke). These timeframes were chosen based on recommendations for the standardised measurement of sensorimotor recovery in stroke trials.[41] Study findings were presented according to these three timeframes.

### Patient and public involvement

There was no patient involvement in this study.

## RESULTS

The initial literature review identified 7589 articles (figure 1). After removing duplicates and screening titles and abstracts, 1083 full text articles were assessed for eligibility. Twenty-eight studies were included in the systematic review.[42–73] Two thousand six-hundred and seventy-seven participants were recruited to these studies—mean participant age was 72.7 years, 49.3% were males and 87% of patients sustained a cerebral infarction. The main reasons for excluding studies were due to not recruiting participants with severe stroke, not providing results separately for participants with severe stroke or not providing sufficient information to determine if the participants had sustained a severe stroke. There was an excellent level of agreement between the two authors in selecting the included articles (Cohen's κ 0.93, percentage of agreement 97.7%).

The characteristics of the included studies are provided in online supplementary file 2 (online supplementary tables 1–3, supplemental references). Sixteen studies

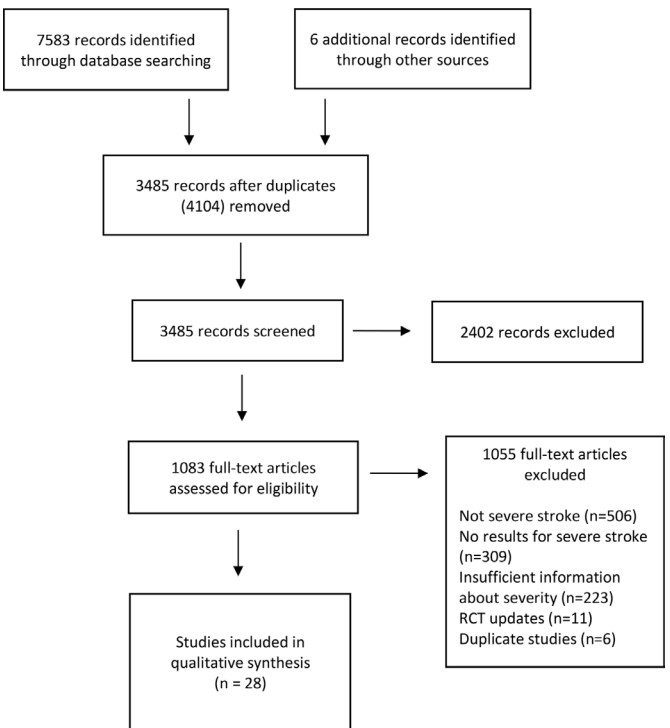

**Figure 1** Flow chart of studies.

were completed within the acute–early subacute phase, eight studies were completed within the acute–late subacute phase and four studies were completed within the chronic phase poststroke. Twenty different interventions were evaluated across the 28 studies. The assessment of risk of bias for each study is presented in figure 2.

### Outcomes

Sixty measures of physical function and immobility-related poststroke complications were identified across the studies. The measures were classified as measures of body function (n=18), activity (n=26), participation (n=8) and poststroke complications (n=8). These measures were grouped together as 16 different outcomes. An overview of these measures and outcomes have been included in online supplementary file 2 (online supplementary table 4).

For each outcome, there was usually only one study investigating the effectiveness of a specific rehabilitation intervention in each time frame poststroke. Most of these studies were rated as providing very low or low-quality evidence for these outcomes (see online supplementary file 2). Outcomes which were supported by studies providing moderate or high quality of evidence are reported in this section. Outcomes which were supported by studies providing low or very low quality of evidence are reported in online supplementary file 2 (online supplementary results, references).

### Body function
#### Sensorimotor function

Seventeen studies evaluated changes in sensorimotor function. Ten studies were completed in the acute to

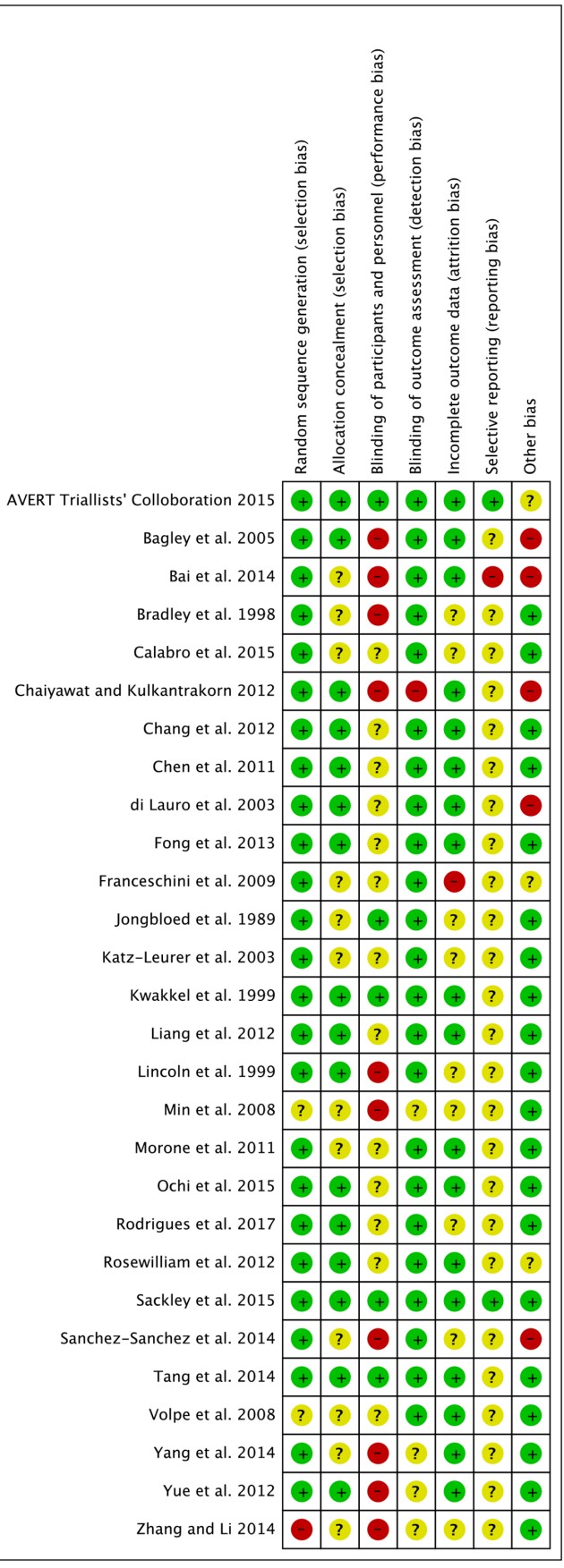

**Figure 2** Risk of bias of individual domains in the included studies.

early subacute phase poststroke,[44–46 48 49 51–55] five studies were completed in the acute to late subacute phase poststroke[59 62 66 68 69] and two studies were completed in the chronic phase poststroke.[70 72] The most frequently used outcome measures of sensorimotor function were the Fugl-Meyer Assessment, used in 11 studies,[45 46 48 51 53 54 59 68–70 72] and the MRC (Medical Research Council) scale for muscle strength, used in five studies.[46 51 52 59 72]

In the acute to early subacute phase poststroke, there was moderate quality evidence from one study that a 6-week course of neuromuscular electrical stimulation (NMES) applied to the wrist and finger extensors in conjunction with usual therapy resulted in no improvement in wrist active movement compared with usual therapy.[55] Wrist strength and grip strength improved in the NMES group during the treatment period although these improvements were not evident at the 9-month follow-up.

## Activity
### Activities of daily living
Twenty studies explored independence and ability to perform ADLs. Eleven studies were completed in the acute to early subacute phase,[42 43 47–49 51–56] seven studies were completed in acute to late subacute phase[58 60–63 66–69] and two studies were completed in the chronic phase.[71 73] Eighteen studies used the BI as the main outcome measure to assess independence in ADLs.[43 47 49 51–53 55 56 58 60–63 66–69 71 73] Four studies used the Modified Rankin Scale[42 49 60 61] and three studies used the FIM.[48 50 54]

In the acute to early subacute phase, there was high-quality evidence that frequent, very early mobilisation (median of 6.5 times per day) commencing within 24 hours poststroke did not result in more patients being less dependent in ADLs at 3 months poststroke compared with usual care, which traditionally started more than 24 hours poststroke and averaged three times per day.[42] However, caution is required with interpreting this finding as the subgroup analysis of patients with severe stroke was not powered for this outcome. There was moderate quality evidence that a 6-week course of NMES applied to the wrist and finger extensors in conjunction with usual therapy resulted in no difference in ADL independence compared with usual care.[55]

In the acute to late subacute phase, there was moderate quality evidence that additional lower limb (LL) therapy in conjunction with regular physical rehabilitation performed in the first 20 weeks poststroke improved ADL independence while the intervention was being delivered when compared with regular physical rehabilitation alone.[63] However, these improvements were not seen 6 months poststroke.

In the chronic phase, there was high-quality evidence that a 3-month occupational therapy (OT) intervention provided to residents in care homes resulted in no difference in ADL independence compared with usual care.[71] Similar caution is required with interpreting this finding

as the subgroup analysis of patients who were severely or very severely disabled was not powered for this outcome.

### Gait
Nine studies investigated gait, which included gait ability and gait speed. Six studies were performed in the acute to early subacute phase,[44–46 49 51 54] two studies were performed in the acute to late subacute phase[63 66] and one study was performed in the chronic phase.[70] The Functional Ambulation Classification was used in eight studies, making it the most frequently used outcome measure of gait ability.[45 46 49 51 54 63 66 70] The 10-m walk test was used in five studies, making it the most frequently used outcome measure of gait speed.[44 49 54 66 70]

Only one study demonstrated moderate quality evidence.[63] In the acute to late subacute phase, additional LL therapy in conjunction with regular physical rehabilitation performed in the first 20 weeks poststroke improved gait ability and speed when compared with regular physical rehabilitation alone. However, these improvements were not seen 6 months poststroke.

### General physical activity
Eight studies examined the effects of different interventions on improving general physical activity. Six studies were performed in the acute to early subacute phase,[43 44 46 51 52 57] one study was performed in the acute to late subacute phase[66] and one study was performed in the chronic phase.[71] General physical activity was defined as a composite of multiple physical tasks completed within one assessment, such as upper limb (UL) or LL function, transfers, gait and balance. Outcome measures used to assess general physical activity included the Rivermead Mobility Index, Rivermead Mobility Assessment and Motor Assessment Scale. Only one study demonstrated high-quality evidence.[71] In the chronic phase, a 3 month OT intervention provided to residents in care homes resulted in no difference in physical activity compared with usual care.

### UL function
Four studies investigated changes in UL function,[52 55 63 72] of which two provided moderate quality evidence.[55 63] In the acute to early subacute phase, a 6-week course of NMES applied to the wrist and finger extensors in conjunction with usual therapy resulted in no difference in UL function compared with usual care.[55] In the acute to late subacute phase, additional UL or LL therapy in conjunction with regular physical rehabilitation performed in the first 20 weeks poststroke improved UL function 6 months poststroke when compared with regular rehabilitation.[63]

## Participation
### Instrumental ADLs
Five studies investigated the effect of different interventions on instrumental ADLs,[43 44 50 62 63] of which one provided moderate quality evidence. Instrumental ADLs are those activities that enable an individual to live independently within their community. In the acute to late

subacute phase, additional UL or LL therapy in conjunction with regular physical rehabilitation performed in the first 20 weeks poststroke improved performance in instrumental ADLs 6 months poststroke when compared with regular rehabilitation.[63]

### Quality of life

Three studies examined quality of life,[60 63 71] of which two were moderate or high quality.[63 71] In the acute to late subacute phase, there was moderate quality evidence that there was no benefit of additional UL or LL therapy to regular physical rehabilitation performed in the first 20 weeks poststroke on improving quality of life 6 months poststroke.[63] In the chronic phase, there was high-quality evidence that a 3-month OT intervention provided to residents in care homes resulted in no difference in quality of life compared with usual care.[71]

### Complications
#### Depression

Four studies explored changes in depression,[43 61 71 72] of which one was high quality.[71] In the chronic phase, a 3-month OT intervention provided to residents in care homes resulted in no difference in depression compared with usual care.[71]

#### Mortality

One study investigated the effect of very early mobilisation on mortality.[42] There was high-quality evidence that frequent, higher dose, very early mobilisation commencing within 24 hours poststroke did not result in more patients dying at 3 months when compared with usual care, which traditionally started more than 24 hours poststroke.

### Other outcomes

There was low quality of evidence for cardiorespiratory function (two studies)[45 49] and caregiver burden (one study).[43] There was very low to low quality of evidence for neurological impairment (three studies),[47 66 68] balance and postural control (eight studies),[43 46 51 57 59 66 70 73] perceived health status (two studies),[43 72] shoulder pain and dislocation (one study),[72] and spasticity (six studies).[44 49 52 58 66 72] Further details of these outcome and studies are included in online supplementary file 2.

### DISCUSSION
#### Main findings

Although 28 RCTs investigating 20 different rehabilitation interventions were identified in this review, there was a paucity of high-quality evidence to support the use of these interventions to improve physical function and reduce immobility-related complications after severe stroke. Most studies were rated as low or very low-quality evidence due to unclear or high risk of bias as well as recruitment of very small samples (refer to online supplementary table 1). However, compared with data from national (United Kingdom) and global estimates of stroke

incidence and prevalence, participants recruited to these studies were similar in terms of stroke type and gender but slightly younger (median age of stroke in the United Kingdom is 77 years).[1 2 18] Therefore, participants were generally representative of the wider stroke population.

### Physical function

Two large, multi-centre studies provided high-quality evidence that their respective treatment interventions were no more effective at improving different aspect of physical function than usual care.[42 71] However, patients with severe stroke or severe disability poststroke comprised a smaller sample within these larger trials. Analyses of data from these subgroups may not be powered to detect changes between the treatment and usual care interventions and therefore caution is required in interpreting the studies' findings.

In A Very Early Rehabilitation Trial (AVERT),[42] very early and frequent mobilisation commencing within 24 hours poststroke did not result in more patients being less dependent in ADLs 3 months poststroke compared with usual care, which traditionally started more than 24 hours poststroke. Although the data seemed to favour usual care practice for patients with severe stroke, this finding did not achieve statistical significance. It could be argued that patients with severe stroke may be less likely to tolerate very early and intensive therapy in the first few days after stroke due to fatigue and reduced exercises tolerance.[74] This would suggest that mobilising patients less intensively after 24 hours may be more beneficial at improving functional recovery than very early and frequent mobilisation. However, this finding was not seen in AVERT.

In the OT in care home trial,[71] a 3-month, goal-orientated OT intervention for stroke survivors living in care homes did not result in improved ADL ability or quality of life up to 1-year postintervention. The authors hypothesised that the lack of treatment effect may have been due to the care home residents' disability severity, which may have limited their engagement in therapy. However, a content analysis of the OT intervention by the research team revealed that the mean number of OT visits over the period was 5.1 (SD 3.0), the median session time was 30 min (IQR 15–60 min) and only 15% of OT time was used to provide ADL and mobility training. Although session length and duration were dependent on the care home resident's ability to engage, it is possible that a more frequent OT intervention that focused more on ADL and mobility training may have resulted in different findings.

Two additional studies provided moderate quality evidence that their respective treatment interventions were effective at improving different aspects of physical function. In both studies, improvements were seen in different aspects of physical function that were specifically trained with the treatment intervention. Kwakkel et al demonstrated that, compared with usual care, a 20-week course of additional UL therapy resulted in improvements in UL function and additional LL training

resulted in improvements in UL function, independence in ADLs, gait speed and gait independence.[63] However, these improvements were not maintained after 6 months poststroke once the additional therapy had discontinued.[64] Rosewilliam *et al* demonstrated that the addition of wrist and finger NMES to usual therapy care resulted in improvements in wrist extensor and grip strength but no difference in UL function nor independence in ADLs.[55] As the electrical stimulation provided to patients was limited to cyclical movements of the wrist and did not involve multiple limb segments, it seems reasonable that UL function and independence in ADLs, which were not specifically trained for with the neuromuscular stimulation, did not improve.

## Immobility-related complications

As demonstrated in online supplementary table 2, there were relatively fewer complication outcomes investigated across all studies compared with physical function outcomes. This observation may reflect that the primary focus of stroke rehabilitation is to optimise functional recovery.[3–5] Therefore, the primary focus of stroke rehabilitation research investigating the effectiveness of rehabilitation interventions may be on improving functional recovery poststroke rather than reducing immobility-related complications.

Only two high-quality studies investigated the effectiveness of their respective interventions at reducing immobility-related complications. In AVERT, very early and frequent mobilisation commencing within 24 hours poststroke did not result in more patients dying at 3 months poststroke compared with usual care.[42] While this finding is obviously positive, very early and frequent mobilisation did not result in less patient dependency as reported earlier in the discussion. Therefore, the optimal time and frequency to commence the mobilisation of patients with severe stroke are not clear.

In the OT in care home trial,[71] a 3-month, goal-orientated OT intervention for stroke survivors living in care homes did not result in reduced depression up to 1-year postintervention. While poststroke depression has a multi-factorial cause, it has been reported that mental distress associated with residual disability may contribute to the development of poststroke depression.[75] Therefore, reductions in residual disability may alleviate depressive symptoms poststroke. As the OT intervention did not result in improved ADL ability, it is possible that depression did not significantly change due to the lack of improvement in ADL ability.

## Implications for practice and research

In light of these findings, it may be necessary to re-evaluate the design of future trials investigating rehabilitation interventions in severe stroke. As it is not known if survivors of severe stroke respond to interventions in the same ways as survivors of milder stroke, there may be a need for more proof of concept studies to understand the mechanisms of recovery in severe stroke more fully. The high number of small, low-quality, single-centre RCTs investigating a broad range of interventions may suggest that larger, high-quality multi-centre RCTs investigating fewer interventions are warranted. However, outcome evaluations alone are insufficient to understand why certain interventions do or do not work. It is recommended that evaluations of complex interventions, such as stroke rehabilitation, use process evaluations alongside outcome evaluations.[76] Process evaluations enable an understanding of how to implement an intervention as well as how participants respond to and interact with the intervention. Therefore, future trials should be guided by more proof of concept research and involve both outcome and process evaluations.

In this review, the most frequently investigated outcomes were functional tasks, such as ADLs and gait ability. However, Pereira *et al* have suggested that individuals with severe stroke are likely to make limited functional improvement with inpatient rehabilitation in their review of rehabilitation after severe stroke.[31] They also advocated more focus on discharge planning and reducing poststroke complications during inpatient rehabilitation for patients with severe stroke. While the extent to which patients can improve functionally after severe stroke is not clear, there is merit in further exploring the effect of rehabilitation in the prevention and management of poststroke complications in severe stroke. Sackley *et al* investigated the prevalence of immobility-related complications in the first year after severely disabling stroke and found a very high prevalence of falls, contractures, pain and pressure sores.[28] However, with the exception of spasticity, there was very little focus on the prevention or management of poststroke complications in the studies selected for our systematic review. In addition to a lack of focus on immobility-related complications, only one study explored caregiver burden, known to be very high among carers looking after survivors of severe stroke.[30] Future research in the rehabilitation of severe stroke should therefore focus more on the effectiveness of rehabilitation interventions in the prevention and management of immobility-related complications in severe stroke.

This review identified several studies investigating technological interventions, such as treadmill training and robot-assistive devices, and more novel interventions, such as thermal stimulation. However, it is not clear how commonly used these interventions are in clinical practice. Additionally, there were no trials studies of interventions commonly used with survivors of severe stroke, such as positioning, sitting balance and seating.[77] This mismatch between available research evidence, which may not reflect current practice, and clinical practice, which may have limited research evidence to support its use, may present a dilemma for therapists, who are expected to base healthcare decisions on the best available and relevant evidence.[78] Therefore, future research is required to understand what interventions are currently being used in clinical practice. Knowledge of currently used rehabilitation interventions may guide future trials investigating

their efficacy in improving physical function and reducing immobility-related poststroke complications.

## Strengths and limitations

In terms of strengths, this is the first systematic review to investigate rehabilitation interventions specifically to survivors of severe stroke, who tend to be under-represented in stroke rehabilitation research, and the identification of topics for future rehabilitation research will hopefully guide much needed research for this cohort of the stroke population. As well, the outcomes of the review focused on not just physical function but immobility-related poststroke complications, which are known to be higher in the severe stroke population and contribute to high levels of caregiver burden.[28–30] In terms of limitations, it has been reported that the defining severe stroke is difficult due to different criteria used to classify severity.[79] The use of objective scores on validated outcome measures to classify stroke severity in our systematic review was deemed necessary to ensure that participants had actually sustained a severe stroke. In our review, the BI was the most commonly used measure to classify stroke severity, reported in 17 out of 28 studies. Using a prespecified score on the BI to classify severe stroke ($\leq$9/20 or $\leq$45/100)[33] enabled the identification of patients with severely disabling stroke. However, the use of an alternative measure of stroke severity, such as the NIHSS, may have resulted in the inclusion of a study with participants with a slightly different clinical presentation than participants measured with the BI. Alternatively, we may have excluded studies that used a different scoring system to classify stroke severity. However, these studies were discussed in detail among three review authors to determine suitability for inclusion and therefore it is likely that the number of relevant studies excluded from the review was minimal. Another limitation is the use of data from subgroups within larger clinical trials. As subgroup analyses may not be powered to detect changes between groups, caution is required in the interpretation of findings from these trials.

## CONCLUSION

There was a paucity of high-quality evidence to support the use of rehabilitation interventions to improve physical function and reduced immobility-related complications after severe stroke. Two high-quality studies suggested that very early mobilisation and OT in care homes were no more effective than usual care. One moderate quality study supported wrist and finger NMES in improving wrist extensor and grip strength. One moderate quality study supported that use of additional UL training on improving UL function and additional LL training on improving UL function, independence in ADLs, gait speed and gait independence. Future research should be guided by more proof of concept studies and involve outcome and process evaluations to more fully understand the impact of different interventions on patients with severe stroke. Future research should investigate the effect of more clinically used interventions, such as positioning, sitting balance and seating. Future research should also investigate the effect of interventions on post-stroke complications known to be high after severe stroke, such as contracture, pressure sores and caregiver burden.

**Contributors** MPMG is the guarantor of the review. MPMG, CS and CMK were involved in the design of the protocol and systematic review. MPMG conducted scoping searches. MPMG and JJ piloted the inclusion/exclusion form. MPMG piloted the data extraction form. MPMG was the first reviewer and JJ was the second reviewer for the systematic review. AD provided statistical support for the systematic review. MPMG drafted the manuscript. All authors read and approved the final manuscript.

**Funding** This project forms part of MPMG's PhD which is funded by The Dunhill Medical Trust (grant number RT62/0116). The funder has had no input on the design of the protocol and will have no input on the analysis and interpretation of the results of the systematic review, or publication of the systematic review.

**Competing interests** None declared.

**Patient consent for publication** Not required.

**Provenance and peer review** Not commissioned; externally peer reviewed.

**Data availability statement** All data relevant to the study are included in the article or uploaded as supplementary information.

**ORCID iD**
Mark P McGlinchey http://orcid.org/0000-0001-5310-2308

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
