## [Reviewer comments · BMJ Open]

ARTICLE DETAILS

TITLE (PROVISIONAL)	The effect of rehabilitation interventions on physical function and immobility-related complications in severe stroke- a systematic review
AUTHORS	McGlinchey, Mark P; James, Jimmy; McKeivitt, Christopher; Douiri, Abdel; Sackley, Catherine

VERSION 1 – REVIEW

REVIEWER	Aurélien Hugues Service de médecine physique et réadaptation, hôpital Henry Gabrielle, Hospices Civils de Lyon, Saint-Genis-Laval, France; Plate-forme “Mouvement et Handicap”, hôpital Henry Gabrielle, Hospices Civils de Lyon, Saint-Genis-Laval, France; Equipe “ImpAct”, Centre de Recherche en Neurosciences de Lyon, Inserm UMR-S 1028, CNRS UMR 5292, Université de Lyon, Université Lyon 1, Bron, France
REVIEW RETURNED	30-Sep-2019

GENERAL COMMENTS	Comments This publication aims at comparing the effectiveness of rehabilitation interventions to another one or usual cares in a specific category of survivors of stroke. The issue of the effectiveness of interventions in survivors of severe stroke is relevant owing to the major consequences for patients, caregivers and society. Abstract The abbreviation “OT” is used but never explained in abstract. “Randomised controlled trials comparing the effect of one type of rehabilitation intervention to another or usual care on physical function and immobility-related complications for patients with severe stroke were included.” What about no intervention as comparator group. The authors have written in protocol published: “The comparator will be any of the following: another type of rehabilitation intervention, usual care or no intervention.” Introduction Background is well described. Introduction is well argued. The objective of study is relevant. Methods Beside the protocol published, it may be interesting to report briefly PICO components in order to make reading easier. Search strategy is well done. Were unpublished and unregistered studies specifically searched? How was resolved conflicts between MM and JJ for data extraction?
---

	A comma is missing between “incomplete outcome data” and “selective outcome reporting”. For strength of evidence: why only factors that may decrease the quality level of a body of evidence (Handbook Cochrane) have been considered? PRISMA statements are followed. Protocol published previously. Methods reported in manuscript submitted are such as declared in Prospero and the protocol published. Results For selection of studies: Cohen’s kappa coefficient could be reported to show the rate of agreement/disagreement between authors in charge of selection of studies. With Revman, forest plots could be showed with risk of bias. Or quality of evidence. The first abbreviation “LL” is not explained (page10-line 12). Discussion “However, there was a trend in the data towards favouring usual care practice for patients with severe stroke.” It seems to me too vague. I am not sure to understand the meaning. “It could be argued that patients with severe stroke, [...], may be less likely tolerate very early and intensive therapy in the first few days after stroke.” Could you justify or explain this statement? In “main findings” and after a long part of results, it could be relevant to summarise main results answering to the question of study. Other parts of discussion are correct. Conclusion Good. This relevance and the quality of the publication is high.
--	--

REVIEWER	Kenneth Lo Flinders University, College of Medicine and Public Health, Australia
REVIEW RETURNED	13-Nov-2019

GENERAL COMMENTS	The review aims to examine the effectiveness of rehabilitation interventions for severely impaired stroke patients, which is a challenging but pertinent topic. There is a range of severe impairment levels across inpatients and outpatients (from very acute phase immediately post stroke to chronic severe impairments), and there are diverse types of rehabilitation interventions, therapy dosages and outcome measures. In view of these various aspects, it would be interesting to examine the effectiveness of rehabilitation therapies for this high-need patient group. Please find below my review comments. page 2:  - line 34: could the full text for 'OT' be provided? - line 41: could the author clarify the statement "additional lower limb on improving upper limb function...?" - line 43: are "independence in activities of daily living" and "independence" two different outcomes? page 3, line 11: are outcomes of physical function included in the review?
--

	page 4, line 36: could the authors provide details on the type of rehabilitation interventions that are commonly prescribed for inpatients and outpatients with severe stroke? This will provide context to the subsequent review findings. page 4 line 42: could the authors recheck the statement for correctness: "Survivors of severe stroke spend longer in hospital..." page 5, Methods:  - can the PICO inclusion criteria for the review be included? Are there any deviations from the protocol PICO? - what are the intervention types, outcomes and associated measurement scales investigated as part of the review? Could the authors elaborate these under the PICO section? page 7, line 22: could the authors provide the "GRADE Summary of Findings" table? page 7, line 28: how were the studies determined as very low, low, medium and high quality? page 7, line 35: could the authors clarify the rationale to conduct meta-analysis only if there are more than 5 studies? page 7, line 41: could the authors elaborate on the nature of the heterogeneity encountered? page 7, line 52: what is there an overlap in terms of timeframe post stroke for studies that are "acute to early subacute phase (0 to 3 months)" and "acute to late subacute phase (0 to 6 months)"? page 8, line 25: would the authors consider conducting meta-analysis based on sub-groups of types of intervention? Looking at the studies in Supplementary Table 1, would it be possible to group these studies into sub-groups of robotics, conventional therapy, functional electrical stimulation, acupuncture, etc? page 8, line 32:  - in view of the review objective, would it be possible to categorise the findings of these 60 measures under: physical function (upper limb, lower limb) and immobility-related complications? Please also see prior comment on the review PICO. - are there specific measures which are more indicative and relevant for upper limb physical function, lower limb physical function, and immobility-related complications? page 9, line 22:  - 11 studies have Fugl-Myer Assessment as an outcome measure, could sub-group meta-analysis based on types of intervention and acute/sub-acute/chronic phases be conducted? - could references be provided for the studies with FM, MRC measures? page 9, line 27: what are the findings for late sub-acute and chronic studies? page 9, line 51:  - there are 18 studies which have BI as an outcome measure, could sub-group meta-analysis be conducted for these studies?
--	---

	- could references be provided for the studies with BI, mRS and FIM? page 10, line 12: could the full words of the abbreviations LL/UL be provided in the main text, where it first occurs? page 10, line 40: - there are 8 studies which have FAC as an outcome measure, could sub-group meta-analysis be conducted for these studies? - could references be provided to identify the studies with FAC, 10m walk test? page 10, line 47: could the reference be provided for the study with moderate quality evidence? page 11, line 18: is reference 65 for the study with high quality evidence? page 10, line 25: what are the outcome measures for upper limb function? page 11, line 43; - for readers' clarity, could the authors provide some background regarding the differences between extended ADL and the prior ADL outcomes? Please also see prior comments on PICO. - what are the outcome scale measures for extended ADL? page 11, line 56: what are the measurement scales for QoL? page 12, line 14 Complications: please describe the outcome scales for 'Depression', 'Mortality', and 'Other Outcomes'. page 12, line 29: - could the dosage of the early mobilisation intervention be provided? - besides mortality rate, are there other outcomes that were examined in the study? page 12, Discussion: - could the discussion points be organised based on the outcomes presented under 'Results'? This would facilitate interpretation of the result findings. - there have been systematic reviews conducted to examine the effectiveness of various rehabilitation interventions (such as functional electrical stimulation, robotics, constraint-induced therapy, task-specific training, treadmill training, etc) for stroke patients. Could the authors take into account the findings of these reviews in their 'Discussion' and 'Implications for Practice and Research'? - could the authors also discuss the type of effective/ineffective interventions for severe patients at various phases of stroke: acute, sub-acute and chronic, in respect of improving physical function and reducing immobility-related complications? page 13, lines 3 and 5: could the references of these 'high risk of bias' studies, small sample size studies, and studies with single centre RCTs be provided? page 13, line 5: could the reasons why these single centre RCTs had less generalisability to wider practice be elaborated?
--	--

	line 9: what are the types of treatment intervention and outcome measures of the two large multicentre studies? Could the authors elaborate further? page 2, line 34: could the full text for 'AVERT' be provided? page 13, line 24: - could the authors elaborate further about the effectiveness of early mobilisation intervention? For severely impaired in-patients, the adoption of early mobilisation (when and in what form) is still unclear, and it would be interesting to understand its effectiveness. Details of effective mobilisation interventions should be provided, such as nature/type of the mobilisation intervention, when to commence early mobilisation, frequency and dosages. Details of the control usual care intervention should also be described in order to understand why these early mobilisation interventions are more effective. - what is the nature of the data and why is there a trend towards favouring usual care practice for patients with severe stroke? page 13, line 27: could the sentence be improved for clarity: "may be less likely tolerate...". page 13, line 31: could the authors clarify this statement in view of the mortality result on page 12, line 25? page 13, line 33: - for the OT in care home trial [65], what are the severity characteristics of the patients living in care homes? - compared to severe patients of the AVERT early mobilisation trial, are there differences in terms of the severity levels and patient characteristics? - how does discussion about the AVERT and OT in care home studies relate back to the review objective? page 15, line 36: In terms of the inclusion/exclusion of studies with severe stroke patients, could the authors discuss further about the nature and range of outcome measures that were encountered during the review, and how do these range of outcome measures compare to the severe range as stated in the protocol.
--	---

VERSION 1 – AUTHOR RESPONSE

Reviewer 1 Comments

Abstract

The abbreviation "OT" is used but never explained in abstract. **I have re-phrased this term in the Abstract.**

"Randomised controlled trials comparing the effect of one type of rehabilitation intervention to another or usual care on physical function and immobility-related complications for patients with severe stroke were included." What about no intervention as comparator group. The authors have written in protocol published: "The comparator will be any of the following: another type of rehabilitation intervention, usual care or no intervention." **I have added "no intervention" as a comparator group in line with the protocol and our systematic review process.**

Methods

Beside the protocol published, it may be interesting to report briefly PICO components in order to make reading easier. **The PICO components were not originally included due to word count limitations. However, I have included a brief summary of PICO.**

Were unpublished and unregistered studies specifically searched? **Ongoing studies not published were identified by searching the Stroke Trials Registry, and this is included already in the text. However, studies need to have been completed in order to obtain study results for analysis.**

How was resolved conflicts between MM and JJ for data extraction? **Conflict was resolved by a third reviewer, which was previously stated in the Methods section. I have altered the text within the Methods section to reflect this.**

A comma is missing between “incomplete outcome data” and “selective outcome reporting”. **I have included the missing comma.**

For strength of evidence: why only factors that may decrease the quality level of a body of evidence (Handbook Cochrane) have been considered? **The GRADE approach starts with the assumption that an article is high quality and then seeks reasons to decrease the quality of evidence by considering the identified factors. It is very rare to increase the quality level of a body of evidence from a randomised controlled trial that has been downgraded and this approach is not commonly performed in stroke rehabilitation systematic reviews. Therefore, we followed the conventional GRADE approach in stroke rehabilitation research.**

Results

For selection of studies: Cohen’s kappa coefficient could be reported to show the rate of agreement/disagreement between authors in charge of selection of studies. **I have calculated Cohen’s kappa and include the results in the Results section.**

With Revman, forest plots could be showed with risk of bias. Or quality of evidence. **Risk of bias is reported in Figure 2 and quality of evidence is reported within Supplementary Table 1.**

The first abbreviation “LL” is not explained (page10-line 12). **I have explained this abbreviation.**

Discussion

“However, there was a trend in the data towards favouring usual care practice for patients with severe stroke.” It seems to me too vague. I am not sure to understand the meaning. **I have rephrased this statement.**

“It could be argued that patients with severe stroke, [...], may be less likely tolerate very early and intensive therapy in the first few days after stroke.” Could you justify or explain this statement? **I have provided a reference to justify this statement.**

In “main findings” and after a long part of results, it could be relevant to summarise main results answering to the question of study. **I have added a summary of the main results answering the study aims in the Conclusion section.**

Reviewer 2 Comments

Please find below my review comments.

page 2:

- line 34: could the full text for 'OT' be provided? **The full text for “OT” has been provided.**

- line 41: could the author clarify the statement "additional lower limb on improving upper limb function..."? **I have added a missing word (“training”) to clarify this statement.**

- line 43: are "independence in activities of daily living" and "independence" two different outcomes? **The second “independence” refers to gait independence. I have clarified this in the text.**

page 3, line 11: are outcomes of physical function included in the review? **Outcomes of physical function are included in the review and the aim of this statement was to highlight a strength of**

this study by including post-stroke complications, which are less commonly reported in stroke rehabilitation research. I have clarified the statement to reflect this.

page 4, line 36: could the authors provide details on the type of rehabilitation interventions that are commonly prescribed for inpatients and outpatients with severe stroke? This will provide context to the subsequent review findings. **There is no current documented evidence describing the types of rehabilitation interventions commonly prescribed to severe stroke patients, which is a limitation of the existing research and forms part of my current research agenda.**

page 4 line 42: could the authors recheck the statement for correctness: "Survivors of severe stroke pend longer in hospital..." **I have corrected this statement by changing "pend" to "spend", which was a spelling error.**

page 5, Methods:

- can the PICO inclusion criteria for the review be included? Are there any deviations from the protocol PICO? **The PICO components were not originally included due to word count limitations, but I have included a brief PICO summary. There were no deviations from the PICO protocol, and I have added a statement to reflect this.**

- what are the intervention types, outcomes and associated measurement scales investigated as part of the review? Could the authors elaborate these under the PICO section? **I have included examples of interventions types, outcomes and measurement scales in this brief PICO summary. Further details are provided in the systematic review protocol.**

page 7, line 22: could the authors provide the "GRADE Summary of Findings" table? **A GRADE Summary of Findings of table is provided to demonstrate the grade of the evidence per outcome. As there were 16 different outcomes and usually only one study per outcome, we presented the grading of evidence for each study within the supplementary table instead of a separate GRADE Summary of Findings table.**

page 7, line 28: how were the studies determined as very low, low, medium and high quality? **I have clarified how the quality of studies were determined within the text.**

page 7, line 35: could the authors clarify the rationale to conduct meta-analysis only if there are more than 5 studies? **Whilst a meta-analysis can technically be conducted with at least 2**

homogenous studies, in order to account for publication bias, which is recognised threat to the

validity of meta-analysis, it is suggested to include at least 5 homogenous studies in meta-analysis.

page 7, line 41: could the authors elaborate on the nature of the heterogeneity encountered? **I have elaborated on the types of heterogeneity encountered.**

page 7, line 52: what is there an overlap in terms of timeframe post stroke for studies that are "acute to early subacute phase (0 to 3 months)" and "acute to late subacute phase (0 to 6 months)"? **I don't fully understand this comment. The time frames have been chosen to reflect the study duration. Some studies were fully completed in the first 3 months post-stroke and some studies were fully completed in the first 6 months post-stroke. This has been documented in the text.**

page 8, line 25: would the authors consider conducting meta-analysis based on sub-groups of types of intervention? Looking at the studies in Supplementary Table 1, would it be possible to group these studies into sub-groups of robotics, conventional therapy, functional electrical stimulation, acupuncture, etc? **In order to provide clinically meaningful findings, it was agreed by all authors that studies could not be grouped together due to marked heterogeneity within the type, dosage, method of delivery and timing of interventions across the different phases post-**

stroke. It was agreed that providing a descriptive summary of the studies would provide more clinically meaningful information to guide clinicians.

page 8, line 32:

- in view of the review objective, would it be possible to categorise the findings of these 60 measures under: physical function (upper limb, lower limb) and immobility-related complications? Please also see prior comment on the review PICO. **In the PICO summary, I have explained the categorisation of physical function according to the International Classification of Function, which divides function into body function, activity and participation. This is a commonly used categorisation in stroke rehabilitation research. The results for physical function are therefore presented according to this definition.**

- are there specific measures which are more indicative and relevant for upper limb physical function, lower limb physical function, and immobility-related complications? **The measures of physical function and immobility-related complications included are those that have been derived from the studies and categorised as above. The 60 measures have been included in Table 2.**

page 9, line 22:

- 11 studies have Fugl-Myer Assessment as an outcome measure, could sub-group meta-analysis based on types of intervention and acute/sub-acute/chronic phases be conducted? **Please see the response to page 8, line 25.**

- could references be provided for the studies with FM, MRC measures? **References have been provided.**

page 9, line 27: what are the findings for late sub-acute and chronic studies? **Only findings from studies with high or moderate quality evidence were provided in the main text. Findings from studies with low and very low quality evidence were provided as a supplementary file. I have added a comment within the main text to reflect this.**

page 9, line 51:

- there are 18 studies which have BI as an outcome measure, could sub-group meta-analysis be conducted for these studies? **Please see the response to page 8, line 25.**

- could references be provided for the studies with BI, mRS and FIM? **References have been provided.**

page 10, line 12: could the full words of the abbreviations LL/UL be provided in the main text, where it first occurs? **I have provided the full words for "LL" and "UL".**

page 10, line 40:

- there are 8 studies which have FAC as an outcome measure, could sub-group meta-analysis be conducted for these studies? **Please see the response to page 8, line 25.**

- could references be provided to identify the studies with FAC, 10m walk test? **References have been provided.**

page 10, line 47: could the reference be provided for the study with moderate quality evidence? **The reference was provided later in the paragraph, but I have changed its position for clarity.**

page 11, line 18: is reference 65 for the study with high quality evidence? **The reference was provided later in the paragraph, but I have changed its position for clarity.**

page 10, line 25: what are the outcome measures for upper limb function? **The outcome measures have been included in Table 2.**

page 11, line 43;

- for readers' clarity, could the authors provide some background regarding the differences between

extended ADL and the prior ADL outcomes? Please also see prior comments on PICO. **I have provided information regarding extended/instrumental ADLs.**

- what are the outcome scale measures for extended ADL? **The outcome measures have been included in Table 2.**

page 11, line 56: what are the measurement scales for QoL? **The QoL scales have been included in Table 2.**

page 12, line 14 Complications: please describe the outcome scales for 'Depression', 'Mortality', and 'Other Outcomes'. **The outcome scales have been included in Table 2.**

page 12, line 29:

- could the dosage of the early mobilisation intervention be provided? **I have provided this information earlier in the Results section when it first appeared.**

- besides mortality rate, are there other outcomes that were examined in the study? **The outcomes that were examined in the study have been previously reported in the Results section and appear in Table 2.**

page 12, Discussion:

- could the discussion points be organised based on the outcomes presented under 'Results'? This would facilitate interpretation of the result findings. **The discussion has been presented according to the two main outcomes- physical function and immobility related complications- as per the reviewer's suggestion.**

- there have been systematic reviews conducted to examine the effectiveness of various rehabilitation interventions (such as functional electrical stimulation, robotics, constraint-induced therapy, task-specific training, treadmill training, etc) for stroke patients. Could the authors take into account the findings of these reviews in their 'Discussion' and 'Implications for Practice and Research'? **Whilst there have been many systematic reviews investigating the effectiveness of a variety of rehabilitation interventions in stroke, studies either rarely include patients with stroke or do not report results specifically for stroke, as reported in the Introduction.**

- could the authors also discuss the type of effective/ineffective interventions for severe patients at various phases of stroke: acute, sub-acute and chronic, in respect of improving physical function and reducing immobility-related complications? **Due to the large number of separate interventions (n=20) and the available word limit, it is not feasible to discuss all the different effective and ineffective interventions in the various stages post-stroke, particularly as most of the studies were rated low or very low quality. Therefore, the discussion has focussed on studies with moderate and high-quality evidence at the various stages post-stroke to guide clinicians to deliver interventions that have moderate to high-quality evidence.**

page 13, lines 3 and 5: could the references of these 'high risk of bias' studies, small sample size studies, and studies with single centre RCTs be provided? **I have referenced Table 1 for this statement as this table contains all this information.**

page 13, line 5: could the reasons why these single centre RCTs had less generalisability to wider practice be elaborated? **The discussion has been altered to reflect the reviewer's previous comment and this statement has been removed during the alteration.**

line 9: what are the types of treatment intervention and outcome measures of the two large multicentre studies? Could the authors elaborate further? **The treatment interventions and outcome measures for the two large studies have been reported in the Table 1.**

page 2, line 34: could the full text for 'AVERT' be provided? **I have provided the full text for "AVERT".**

page 13, line 24:

- could the authors elaborate further about the effectiveness of early mobilisation intervention? For severely impaired in-patients, the adoption of early mobilisation (when and in what form) is still unclear, and it would be interesting to understand its effectiveness. Details of effective mobilisation interventions should be provided, such as nature/type of the mobilisation intervention, when to commence early mobilisation, frequency and dosages. Details of the control usual care intervention should also be described in order to understand why these early mobilisation interventions are more effective. **The early mobilisation was not more effective than usual care in terms of reducing dependency nor did it result in greater mortality than usual care. I have included the timing and dosage of early mobilisation versus usual care within the Results section. Some details of the different interventions have also been provided in Table 1.**

- what is the nature of the data and why is there a trend towards favouring usual care practice for patients with severe stroke? **I have clarified this statement within the text.**

page 13, line 27: could the sentence be improved for clarity: "may be less likely tolerate...". **I have clarified this statement within the text.**

page 13, line 31: could the authors clarify this statement in view of the mortality result on page 12, line 25? **I have rephrased this statement in terms of functional recovery.**

page 13, line 33:

- for the OT in care home trial [65], what are the severity characteristics of the patients living in care homes? **The severity characteristics of the OT care home trial have been reported in Table 1.**

- compared to severe patients of the AVERT early mobilisation trial, are there differences in terms of the severity levels and patient characteristics? **Severity was measured using two different outcome measures in these two trials (Barthel Index and NIHSS) and therefore comparison between these two patient cohorts is not possible. I have referred to this observation in the Strengths and Limitations section of the Discussion.**

- how does discussion about the AVERT and OT in care home studies relate back to the review objective? **I have altered the Discussion to discuss each of these studies in relation to the review objective.**

page 15, line 36: In terms of the inclusion/exclusion of studies with severe stroke patients, could the authors discuss further about the nature and range of outcome measures that were encountered during the review, and how do these range of outcome measures compare to the severe range as stated in the protocol. **I have discussed this matter further in the Strengths and Limitations section of the Discussion.**

VERSION 2 – REVIEW

REVIEWER	Aurélien HUGUES Service de médecine physique et réadaptation, hôpital Henry Gabrielle, Hospices Civils de Lyon, Saint-Genis-Laval, France; Equipe "ImpAct", Centre de Recherche en Neurosciences de Lyon, Inserm UMR-S 1028, CNRS UMR 5292, Université de Lyon, Université Lyon 1, Bron, France.
REVIEW RETURNED	16-Dec-2019
GENERAL COMMENTS	Comments Very interesting and relevant work. Clear and well argued. All comments have been taken into account. This work is clearly suitable to publish.

	Results Page 12: "In the acute to late subacute phase, additional LL therapy in conjunction with regular physical rehabilitation performed in the first 20 weeks post-stroke improved gait ability and speed when compared to regular physical rehabilitation alone. However, these improvements were not seen 6 months post-stroke." Warning, " " is empty.
--	--